# Biomarker-Based Analysis of Pain in Patients with Tick-Borne Infections before and after Antibiotic Treatment

**DOI:** 10.3390/antibiotics13080693

**Published:** 2024-07-25

**Authors:** Kunal Garg, Abbie Thoma, Gordana Avramovic, Leona Gilbert, Marc Shawky, Minha Rajput Ray, John Shearer Lambert

**Affiliations:** 1Te?ted Oy, 40100 Jyväskylä, Finland; kunal.garg@tezted.com (K.G.); leona.gilbert@tezted.com (L.G.); 2Department of Infectious Diseases, Catherine Mc Auley Education & Research Centre, Mater Misericordiae University Hospital, 21 Nelson Street, Dublin 7, D07 A8NN Dublin, Ireland; abbie.thoma@ucdconnect.ie (A.T.); gavramovic@mater.ie (G.A.); 3Université de Technologie de Compiègne, Costech Laboratory, Alliance Sorbonne Université, Centre de Recherches, 60203 Compiègne, France; 4Curaidh Clinic: Innovative Solutions for Pain, Chronic Disease and Work Health, Perth PH2 8EH, UK; drminha@curaidh.com; 5Catherine Mc Auley Education & Research Centre, University College Dublin, 21 Nelson Street, Dublin 7, D07 A8NN Dublin, Ireland; 6Infectious Diseases Department, The Rotunda Hospital, D01 P5W9 Dublin, Ireland

**Keywords:** Lyme disease, tick-borne disease, pain, post-treatment Lyme disease syndrome, chronic Lyme disease

## Abstract

Tick-borne illnesses (TBIs), especially those caused by Borrelia, are increasingly prevalent worldwide. These diseases progress through stages of initial localization, early spread, and late dissemination. The final stage often leads to post-treatment Lyme disease syndrome (PTLDS) or chronic Lyme disease (CLD), characterized by persistent and non-specific multisystem symptoms affecting multiple systems, lasting over six months after antibiotic therapy. PTLDS significantly reduces functional ability, with 82–96% of patients experiencing pain, including arthritis, arthralgia, and myalgia. Inflammatory markers like CRP and TNF-alpha indicate ongoing inflammation, but the link between chronic pain and other biomarkers is underexplored. This study examined the relationship between pain and biomarkers in TBI patients from an Irish hospital and their response to antibiotic treatment. Pain ratings significantly decreased after antibiotic treatment, with median pain scores dropping from 7 to 5 (*U* = 27215.50, *p* < 0.001). This suggests a persistent infection responsive to antibiotics. Age and gender did not influence pain ratings before and after treatment. The study found correlations between pain ratings and biomarkers such as transferrin, CD4%, platelets, and neutrophils. However, variations in these biomarkers did not significantly predict pain changes when considering biomarkers outside the study. These findings imply that included biomarkers do not directly predict pain changes, possibly indicating allostatic load in symptom variability among long-term TBI patients. The study emphasizes the need for appropriate antibiotic treatment for TBIs, highlighting human rights issues related to withholding pain relief.

## 1. Introduction

The prevalence of tick-borne infections (TBIs) is rising worldwide, with Borrelia being the main pathogen responsible for TBIs in Europe [1,2]. Borrelia infection progresses through three stages [3,4,5]: Stage 1: Early localized, which starts three days after the tick bite and usually looks like flu-like symptoms; about two-thirds of people who reach this stage develop erythema migrans; Stage 2: Early dissemination, which happens weeks to months after infection; patients may have generalized lymphadenopathy, fatigue, and neurological symptoms like encephalitis and cranial neuritis; Stage 3: Late disseminated, which shows up months to years after infection. Symptoms in Stage 3 are often chronic and nonspecific, spanning multiple systems such as neurological, cardiac, cognitive, and arthritic presentations. TBIs can lead to multisystem disease, ranging from asymptomatic cases to debilitating conditions [6,7,8,9]. While most individuals appear to recover during Stages 1 or 2 with recommended antibiotic therapy, Stage 3 may result in prolonged and non-specific multisystem symptoms that are challenging to treat. It is estimated that 10–27% of patients treated with antibiotics experience prolonged symptoms ranging from fatigue to musculoskeletal pain to cognitive impairment, and this is commonly referred to as post-treatment Lyme disease syndrome (PTLDS) [5,10,11,12,13,14].

Another term often used to describe the persistence of symptoms following treatment for tick-borne infections is “chronic Lyme disease” (CLD), which lacks a precise medical definition and is often used interchangeably with PTLDS, though it can imply a continuous infection rather than a post-infectious syndrome [15,16]. “Chronic multisymptom illness” (CMI) is another term encompassing a broader range of persistent, unexplained symptoms affecting multiple body systems and is sometimes preferred to emphasize the multi-organ impact of these conditions [11,17]. Additionally, “post-acute infection syndrome” (PAIS) describes the lingering symptoms that can occur after the acute phase of an infection, highlighting the potential inadequacies or delays in the initial treatment [18]. These terms underscore the ongoing debate regarding the nature of persistent symptoms, whether they are due to an active infection, an autoimmune response, or other mechanisms. The adequacy, timeliness, and effectiveness of initial treatments are often scrutinized in discussions about these conditions, as these factors can significantly influence patient outcomes [19]. Exploring these various terminologies and their implications can enhance our understanding of the chronic manifestations associated with TBIs.

PTLDS refers to the persistence of symptoms for more than six months despite appropriate antibiotic treatment, which results in functional impairment. Although this condition is found post-initial treatment for TBIs, it is not necessarily post-infectious [20,21,22]. Pain is a commonly reported symptom in PTLDS patients [11,23]. A study of late-stage patients experienced long-lasting arthritis, arthralgia, and myalgia [24,25]. This pain is defined as chronic, according to the International Association for the Study of Pain (IASP), as it persists or recurs for longer than three months [26]. Numerous studies have classified the chronic pain that PTLD patients experience as either neuropathic or nociceptive [27]. Chronic pain is thought to be more common in women than men, with large-scale epidemiological studies consistently showing that women report experiencing pain more frequently than men across various anatomical regions [28]. Women are also more likely to report chronic, widespread pain and have a higher prevalence of common chronic pain conditions such as fibromyalgia and migraine. However, determining whether the severity of pain differs between sexes shows conflicting reports, with some studies suggesting that women experience greater pain severity while others find no differences [28]. Advancing age is associated with an increased risk of chronic pain, and population studies in both the US and globally consistently show a higher prevalence of chronic pain in adults over 65 compared to the general adult population [29,30]. However, increased prevalence does not mean that they experience greater pain intensity, and studies [31,32] that compared the pain intensity of patients with chronic pain across three different age groups suggested that pain intensity does not significantly vary with age.

There have been few studies analyzing biomarkers associated with chronic pain. Inflammatory markers such as pro-inflammatory cytokines such as TNF-alpha and CRP (C reactive protein) can indicate ongoing inflammation and have been significantly higher in PTLDS patients [33,34]. For T-cell counts, the results were inconsistent, with some studies finding no significant differences in the total number of circulating CD4+ and CD8+ T-cell counts between chronic pain patients and pain-free controls and others showing a lower number of CD8+ T cells and a higher CD4+/CD8+ ratio between those with chronic pain and the controls [35]. Previous investigations have linked chronic Lyme disease symptoms to diminished CD57+ natural killer (NK) cell counts, highlighting the importance of exploring additional immune markers for dysfunction [36,37,38]. CD3+, CD4+, and CD8+ T-cell lymphocytes are crucial in pathogen-specific adaptive immune responses [39]. However, chronic conditions can alter memory T-cell differentiation programming, leading to T-cell exhaustion and immune response failure, which could serve as a diagnostic area in patients with chronic symptoms [39]. Previous investigations have linked chronic Lyme disease symptoms to diminished CD57+ natural killer (NK) cell counts, highlighting the importance of exploring additional immune markers for dysfunction [36,37,38].

In addition, recent investigations highlight the role of T cells in mitigating pain and preventing its transition from acute to chronic phases. Various T-cell subsets, including suppressor CD8+ and CD4+ T cells, have been identified as contributors to pain alleviation by promoting an anti-inflammatory environment [35]. Furthermore, CD4+ and CD8+ T cells release endogenous opioids, such as enkephalins and endorphins, which bind to opioid receptors on sensory neurons, thereby diminishing pain sensation [40]. Immune dysfunction resulting from TBIs may lead to T-cell exhaustion, potentially exacerbating patients’ pain symptoms. Monitoring T-cell activity could serve as a valuable approach to symptom management and a target for therapeutic interventions to alleviate chronic pain.

An essential field of research in comprehending chronic pain disorders related to infectious diseases is examining the connection between pain, biomarkers, and infections. Biomarkers are biological markers that can indicate the existence and severity of illnesses. Biomarkers offer an objective metric that can assist in diagnosing and treating pain, a subjective and frequently difficult condition to assess. Here, we investigate the correlation between pain and standard biomarkers utilized at a hospital in a cohort of tick-borne disease patients. Examining abnormalities in these immune markers relative to chronic symptoms and their improvement may unveil characteristic biomarker patterns for TBIs.

## 2. Results

In this study, which involved 186 patients between 17 and 81 years, with a median age of 43 (Figure 1A), we observed no significant influence of age or gender on pain ratings. Our cohort comprised 113 female and 73 male participants (Figure 1B). Scatter plots correlating age with pain ratings overlaid with regression lines for time points T0, and T2 revealed trend lines almost parallel to the *x*-axis, indicating negligible age effects on pain ratings at both time points (Figure 1C). Violin plots, incorporating kernel density estimates for each gender across these time points, also showed that gender does not significantly affect pain ratings. However, a noticeable decrease in pain ratings from T0 to T2 was evident regardless of gender (Figure 1D).

Quantitative analysis of pain intensity, measured on a scale from 1 (low pain) to 10 (severe pain), demonstrated a leftward shift in the pain rating distribution curve from T0 to T2, suggesting a general reduction in pain (Figure 2A). The median pain rating decreased from 7 at T0 to 5 at T2 (*U* = 27215.50, *p* ≤ 0.001). The shift in pain ratings from T0 to T2 has a humpback shape, suggesting that there are early and late responders to treatment. Additionally, the distribution of pain ratings between these time points was statistically significant (K-S statistic = 0.43, *p* ≤ 0.001), as illustrated in Figure 2C.

We also measured 18 biomarkers at time points T0 and T2 for these patients. The overall distributions and changes in median values for each biomarker are depicted in Appendix A. The changes in distribution are displayed using empirical cumulative distribution function (ECDF) plots in Appendix A. Table 1 summarizes these findings, highlighting significant changes in median values for transferrin (*U* = 25815, *p* ≤ 0.001), transferrin saturation % (*U* = 13646, *p* ≤ 0.001), platelets (*U* = 20555, *p* ≤ 0.01), CD4% (*U* = 14399, *p* ≤ 0.01), and neutrophils (*U* = 19944, *p* ≤ 0.05). The overall distributions for the same biomarkers in Table 1 were also significantly different (K-S statistics ranging from 0.15 to 0.40, *p* ≤ 0.05). No significant differences were observed regarding the median or overall distribution for the other 13 markers.

While assessing the association between pain rating and statistically relevant biomarkers from Table 1, Figure 3A and Appendix A reveal that median transferrin levels were consistently lower at T2 across all pain ratings (1 to 10), with particularly significant changes observed from pain levels 6 to 9. Unlike transferrin, the median values from other biomarkers did not exhibit consistent trends between T0 and T2 (Figure 3). For CD4%, statistically significant differences in median values occurred at pain ratings of 3 and 7 (Figure 3B, Appendix A). In the case of platelets, notable changes were observed at pain levels 3 and 10 (Figure 3C, Appendix A). Neutrophils only showed significant median value changes at pain level 7 (Figure 3D, Appendix A). No statistically significant alterations were detected across different pain ratings in the transferrin saturation % (Figure 3E, Appendix A). This mixed pattern suggests a complex interplay between pain levels, these specific biomarkers over time, and pain ratings.

To further illustrate the relationship between transferrin, CD4%, platelets, neutrophils, transferrin saturation %, and pain ratings, we created cluster maps for time points T0 and T2 (Figure 4). At the top of each map, dendrograms cluster the pain ratings alongside the said biomarkers, while dendrograms on the left cluster patients based on their similarities (Figure 4). Blue and red colors on each cluster map indicate lower or higher biomarker values, respectively (Figure 4). Biomarker and pain rating values were standardized to range between 0–1 due to varying scales and units (Figure 4). At time T0, two distinct groups were evident in the top dendrograms shown in Figure 4A. The first group, predominantly blue, included transferrin saturation %, platelets, and neutrophils, indicating lower values (Figure 4A). The second group included pain ratings alongside CD4% and transferrin (Figure 4A). This suggests that at T0, CD4% and transferrin are closely linked to pain ratings. However, these relationships change after treatment (at T2) (Figure 4B). While platelets and neutrophils continue to show association, the link between pain ratings, CD4%, and transferrin becomes distant (Figure 4B). Despite an overall reduction in pain ratings from T0 to T2—as evidenced by less red coloration in the cluster maps—certain patients with high pain ratings continue to show elevated CD4%, as observed in Figure 4B, bottom left. However, in Figure 4B, top left, high CD4% are no longer linked to higher pain ratings. Still, these patients with high CD4% and low pain ratings show high transferrin levels, suggesting that factors other than CD4% and transferrin may influence the reduction in pain ratings.

We also applied the K-nearest neighbors (KNN) method to understand changes in the cluster maps from T0 to T2 (Appendix A). The KNN approach uses Euclidean distances—a straightforward distance measure in space to compare pain ratings with transferrin, CD4%, platelets, neutrophils, and transferrin saturation % (Appendix A). At T0, patients with the highest pain rating of 10 had significantly distinct biomarker profiles, as they were farthest from other patients regarding their high Euclidean distance (Appendix A). Patients reporting the same pain level (with pain on a scale ranging from 1 to 10) had the closest biomarker profiles. Furthermore, patients reporting level 10 pain are distant from all other patient groups. After treatment (T2), however, the distances for patients reporting a pain level of 10 decreased considerably, indicating that their biomarker profiles became similar to those of patients with lower pain ratings. This may indicate a normalization progression (Appendix A).

Furthermore, in Table 2, the regression was performed without including a constant term, implying that the model assumes the dependent variable (i.e., change in pain rating from T0 to T2) has no baseline value when all the predictors are zero. The RLM in Table 2 only measures changes in pain ratings from T0 to T2, including changes in transferrin, CD4%, platelets, neutrophils, and transferrin saturation % from T0 to T2. It does not consider any other factors not measured in the current study. The results indicate that changes in transferrin between the two time points are a significant predictor, with a coefficient of 3.29 (*p* < 0.001). This suggests that for each unit increase or decrease in transferrin, the pain ratings will increase or decrease by approximately 3.29 units (Table 2). Similarly, changes in CD4% have a negative coefficient of −0.12 (*p* < 0.01), indicating that each unit increase in CD4% is associated with a decrease in pain rating by 0.12 units (Table 2). However, other predictors like changes in platelets, neutrophils, and Transferrin saturation % do not show significant associations with changes in pain rating as indicated by their *p*-values in Table 2 (0.199, 0.634, and 0.830, respectively).

Table 3 includes a constant term in the regression model, accounting for a baseline change in pain due to factors outside the study when all predictor variables (i.e., transferrin, CD4%, platelets, neutrophils, and transferrin saturation %) are zero. The constant term coefficient is −2.49 (*p* < 0.001), suggesting that the baseline change in pain is approximately −2.49 units (Table 3). This implies a significant initial decrease in pain change when all predictors are at baseline levels (Table 3). Unlike Table 2, Table 3 shows that none of the predictors—including changes in transferrin, CD4%, platelets, neutrophils, and transferrin saturation %—demonstrate a significant association with changes in pain in addition to the baseline change coefficient of −2.49. Therefore, Table 2 shows that when the RLM model predicts changes in pain using only the study’s biomarkers, transferrin and CD4% significantly affect pain ratings. However, when including unmeasured factors, changes in the five biomarkers do not influence pain ratings beyond the baseline change observed when all predictors are zero (Table 3).

## 3. Discussion

### 3.1. Key Findings

The key findings from the data provided suggest a complex relationship between biomarkers, demographics, and the experience of pain. Firstly, there is no observed correlation between age and gender in pain levels. They indicate that these demographic factors do not significantly influence pain perception or reporting. Secondly, there is a clear indication of improvement in pain scores following treatment, indicating antibiotic treatment’s effectiveness in reducing pain. Moreover, pain seems to shift into 2 group patterns: early responders and late responders. Thirdly, transferrin, CD4%, platelets, neutrophils, and transferrin saturation % demonstrate significant differences in distribution and median values before and after treatment. Although the numerical differences in biomarkers such as transferrin, CD4%, platelets, neutrophils, and transferrin saturation % may appear small, their statistical significance indicates meaningful changes. These biomarkers play critical roles in inflammatory and immune responses, essential in understanding the pathophysiology of TBIs and the response to antibiotic treatment. The observed changes align with previous research demonstrating significant symptom improvement following antibiotic therapy [13,41], highlighting the clinical relevance of these findings. Lastly, when the RLM model predicts pain changes using only the five biomarkers, transferrin and CD4% significantly affect pain ratings. However, including unmeasured factors shows that changes in the five biomarkers do not influence pain ratings beyond the baseline change observed when all predictors are zero. This implies that while these biomarkers may be necessary for other clinical assessments or conditions, they are inadequate indicators of pain levels or the efficacy of pain management in this context. Lastly, patients reporting the same pain level had the closest biomarker profiles. After treatment, however, the distances for patients reporting a pain level of 10 decreased considerably, indicating that their biomarker profiles became similar to those of patients with lower pain ratings. This may indicate a normalization progression.

### 3.2. Age and Gender

Research from Goren et al. [42] and Thompson et al. [43] have demonstrated the need to consider patient demographics while studying vector-borne illness’s seasonal and emerging patterns. Moreover, they have shown that older persons experience a significantly higher rate of Lyme borreliosis, and they suggest increasing public awareness about this vulnerable population. Nevertheless, there is limited knowledge regarding the impact of age and gender on illness patterns among the overall population. Additional studies have emphasized the alterations in immunological markers, disease progression, and pathogenesis in patients with Lyme disease (LD) and chronic Lyme disease (CLD), dependent on their age and sex, as reported by many [8,44,45]. Our study hypothesized that age and gender were risk factors contributing to the development of chronic symptoms associated with pain in our TBI cohort. However, our findings indicate that age and gender do not affect the pain experienced within our study population.

### 3.3. Pain and Biomarkers

Current studies have been investigating many aspects of pain biomarkers. Reliable biomarkers for chronic pain, a common illness that affects a substantial part of adults, still need to be identified [46]. Allen and colleagues [46] have emphasized the difficulty of identifying biological indicators that can reliably forecast or diagnose chronic pain conditions. Other research [47] examined the possibility of creating objective biomarkers to transform how we diagnose and treat neuropathic pain significantly. Their research emphasized the significance of composite biomarkers that accurately represent the intricate pathophysiology of neuropathic pain, hence improving treatment precision [47]. Although we have demonstrated that specific biomarkers such as transferrin, CD4%, platelets, neutrophils, and transferrin saturation % are significantly abnormal in this cohort, their correlation to pain and pain scores is insignificant beyond the baseline change in pain. Inflammatory indicators have been recognized as possible biomarkers for many pain disorders [48], and scientists have presented empirical data demonstrating a direct relationship between pro-inflammatory biomarkers, including CRP, and the intensity of low back pain. The observed link could facilitate the development of specific anti-inflammatory treatments that could provide more efficient pain relief [48]. Similarly, Calapai and co-workers [49] investigated the ability of immune biomarkers, such as CRP, to forecast the severity of pain in cancer patients. They propose that this research could lead to individualized pain treatment approaches that consider individuals’ unique biomarker profiles [49].

In the context of chronic pain, infections have a substantial impact on both the beginning and continuation of pain conditions. The significance of inflammatory and metabolic indicators, such as myeloperoxidase, soluble tumor necrosis factor-alpha receptor II, and CRP, was highlighted in centralized pain syndromes. These biomarkers are indicators of neuroinflammation and metabolic disruptions that might worsen pain, especially in the presence of infections [50]. Building upon this topic, Cohen et al. [51] elaborated on the connection between persistent pain following acute diseases and the involvement of immunological processes and direct microbial invasion. The authors specifically include viruses such as HIV, SARS-CoV-2, Mycobacterium leprae, and Borrelia, emphasizing the intricate mechanisms via which infections might result in persistent pain syndromes [51]. The molecular pathways underlying pain caused by urinary tract infections (UTIs) provided a clear example of how pathogens directly affect the sense of pain. In their study, Rudick et al. [52] investigated the role of Toll-like receptor 4 (TLR4) in the perception of pain in urinary tract infections (UTI). They found that UTI can cause pain, unlike cases of asymptomatic bacteriuria when no pain is reported. This highlights the importance of specific bacterial interactions in triggering pain responses [52].

Another intriguing element is the direct detection of infections by nociceptors, sensory neurons that respond to harmful or potentially harmful stimuli. In their study, Chiu and colleagues [53] specifically investigated the direct detection method, suggesting that nociceptors can identify harmful components without the immune system’s involvement. As a result, this direct recognition leads to the immediate experience of pain [53]. In another study, Rosen and Klumpp [54] extensively investigate the precise bacterial constituents, such as lipopolysaccharides derived from *E. coli*, which interact with receptors and trigger pain in urinary tract infections. According to their findings, these pathways may also play a role in chronic illnesses such as visceral pain and interstitial cystitis [54]. Finally, Wang et al. [55] analyze the capacity of antimicrobial medicines to alleviate pain. These treatments are shown to decrease pain by reducing the amount of infection and interfering with the pathways that communicate pain. Like our results (Figure 2), Wang et al. [55] provided therapeutic benefits for illnesses such as persistent low back pain and irritable bowel syndrome [55,56]. These studies demonstrate the complex relationships between infectious agents, biomarkers, and pain. In addition, our study and others offer crucial observations that efficiently facilitate the development of specific treatments to control pain in the context of infectious disorders.

This lack of correlation between pain and biomarkers, as seen in this study (Table 3), may be due to the allostatic load and the impact that chronic Lyme disease has on this cohort [57]. The concept of the allostatic load has been investigated in pain [58]. The allostatic load model considers several parameters, including age, gender, level of education, smoking habits, alcohol intake, physical activity, depression, and other common health conditions. Research on allostatic load pain models has shown a clear relationship between the severity of pain and the degree of allostatic load [58]. It is hypothesized that allostatic load is responsible for the severity and array of symptoms [41] in this cohort of patients suffering long-term tick-borne infections.

Ultimately, the search for conclusive pain biomarkers is hindered by pain’s subjective character and complex molecular foundations. Ongoing research is essential since comprehending and confirming putative biomarkers can completely transform pain treatment, resulting in more accurate diagnostics and individualized therapy methods. Implementing this would improve the quality of care for patients and potentially reduce the significant societal and economic impact of chronic pain problems.

### 3.4. Implications for Practice & Policy

Patients with TBIs report high levels of pain, as seen in our study. Pain scales range from minimal (1), mild (2), uncomfortable (3), moderate (4), distracting (5), distressing (6), unmanageable (7), intense (8), severe (9) and immobilizing (10) [59]. Our study shows an overall movement for mean pain levels from unmanageable pain, rated at 7 before treatment, to distracting pain, rated at 5 after treatment. This is significant for disability levels as studies have found a significant positive correlation between pain intensity scores and disability levels, with reductions in pain intensity corresponding to improvements in functional ability and disability scores [60]. Shifts in pain patterns suggest two groups of patients: early responders and late responders to treatment. An analysis of the infection length may provide further insight into early and late responders, and further research is needed.

A recent study has shown that various gaslighting behaviors are present among healthcare professionals dealing with patients with chronic Lyme disease and other TBIs. Rather than attending to the symptoms, patients are much more likely to be told by practitioners that they are just overreacting to their symptoms or that their symptoms are caused by normal aging, mental illness, or stress. Many patients also feel that medical professionals frequently imply that their symptoms are merely psychosomatic [61]. Although the relationship between pain and biomarkers is not fully understood, our study shows that the pain symptoms experienced by patients significantly decrease with antibiotic treatment. Moreover, several studies highlight the positive impact of longer antibiotic treatments [62,63,64,65]. This raises an issue about the withholding of antibiotic treatment for those patients. The Universal Declaration of Human Rights (UDHR), adopted by the United Nations General Assembly in 1948, includes the right to health as a fundamental human right. Access to timely and adequate healthcare, including pain management, is essential for realizing this right. Thus, individuals have the right to receive appropriate treatment for pain without discrimination. As treatment consisted of antibiotics, not pain killers or pain blockers, and it was effectively reducing pain in this cohort of patients, our study raises an issue about the human rights of patients in terms of the frequent withholding from treatment reported for chronic Lyme/PTLDS and other TBIs because of scientific debates regarding the nature of this condition about the human rights of patients.

### 3.5. Study Limitations and Future Research Directions

This study has several limitations that should be considered. Firstly, while substantial, the cohort size of 186 patients may differ from the broader population with similar conditions, potentially limiting the generalizability of our findings. Secondly, the reliance on self-reported pain ratings introduces a subjective element that could affect the accuracy of the results. We only measured a specific set of biomarkers, which might not encompass all relevant biological factors influencing pain perception and treatment response. The retrospective and longitudinal nature of the study, while valuable for observing changes over time, does not include a control group, which limits the ability to draw definitive causal conclusions about the treatment’s effectiveness. Expanding the range of biomarkers studied will provide a more comprehensive understanding of the factors influencing pain and treatment outcomes. Investigating the role of other potentially confounding variables, such as psychological factors and the duration of infection, could also yield valuable insights into the complexities of pain management in patients with tick-borne infections who experience ongoing symptoms. Lastly, although participants received various antibiotic regimens with different durations [13], the study focused on the overall trend in pain reduction across the cohort. The significant reduction in pain scores observed in Figure 2 due to the overall treatment regimens suggests a generalizable effect of antibiotic therapy. While the natural course of the disease may lead to spontaneous pain resolution, the statistical significance and consistency of the pain reduction point towards a treatment effect (Figure 2 and Figure 3). While a placebo-controlled group would provide stronger causal evidence, the open-label design still offers valuable insights, supported by our prior studies demonstrating the efficacy of antibiotic treatments in reducing TBI symptoms [13,41].

## 4. Materials and Methods

### 4.1. Ethics Approval

The study obtained ethical approval from the Institutional Review Board of the Mater Misericordiae University Hospital (Institutional Review Board Reference: 1/378/1946). The study adheres to the study protocol (version 6), the EU CT Directive 2001/20/EC, GCP Commission Directive 2005/28/EC, ICH/GCP, the Declaration of Helsinki (1996 Version), and all other relevant local and international regulatory requirements.

### 4.2. Patient Cohort and Sample Size Estimation

This study invited individuals over 16 who obtained consultations at the outpatient infectious diseases clinic at The Mater Misericordiae University Hospital in Ireland. The purpose of the study was to include patients suspected of having tick-borne infections. After being evaluated by the doctor during the initial visit (T0), a total of 301 patients were examined, and 210 were diagnosed with tick-borne infections (TBIs) and prescribed antibiotics from T0 (baseline before treatment) to T2 (after antibiotic treatment). These individuals exhibited symptoms resembling Lyme disease, such as a general flu-like sickness and a medical suspicion of infections transmitted by ticks. This suspicion was based on factors such as one or several tick bites, previous occurrence of a distinctive rash resembling a bull’s eye, or prior exposure to locations where ticks are prevalent. Participants completed a survey [13,19] that assessed their pain levels on a scale of 1 to 10 (with 10 indicating severe pain) before and after treatment, time points T0 and T2. This study analyzed a total of 186 participants in its final cohort. The study employed a longitudinal methodology, which involved studying changes over time within a consistent group of participants. This strategy differs from retrospective case-control studies, which compare people with a given outcome (cases) to individuals without the outcome (controls) [66,67].

The sample size for this study was determined based on Cohen’s *d*-effect size of 0.43, derived from a previously published study on a similar cohort of patients. To ensure sufficient statistical power, we aimed for a power of 0.8 (80%) with a significance level (alpha) set at 0.05. The required sample size was calculated using the following formula:$$n={\left(\frac{{(Z}_{\alpha /2}+Z_{\beta })}{∆/\sqrt{2(1-\rho )}}\right)}^{2}$$
where *n* is the sample size, *Z_α_*_/2_ is the critical value of the normal distribution at *α*/2 (for α = 0.05, *Z_α_*_/2_ = 1.96), *Z_β_* is the critical value of the normal distribution for power 1 − *β* (for *β* = 0.2, *Z_β_* = 0.84), and Δ is the standardized effect size. Assuming a moderate correlation (*ρ* = 0.5), the required sample size per group was calculated to be approximately 85. With 186 participants at each time point, our study is well-powered to detect statistically significant differences over time, providing more robust and reliable results. This larger sample size enhances our ability to observe true effects and minimizes the risk of Type II errors (the probability of failing to detect a true effect). The chosen significance level of 0.05 corresponds to a 5% risk of committing a Type I error, which is the probability of incorrectly rejecting the null hypothesis when it is true. Balancing the risks of Type I and Type II errors is essential to maintain the integrity of the study’s findings. For this calculation, we used the standard formula for determining sample size in paired samples hypothesis testing as detailed in the literature [68].

### 4.3. Analyses of Immune Markers and Biomarkers

Immune indicators and biomarkers were analyzed by collecting blood samples and examining them for abnormalities before (T0) and after treatment (T2). The immunological markers analyzed in this group were CD3+, CD8+, CD4+, CD57+, CD19+, cell percentages, and cell counts. In addition, we conducted tests to determine the H/S ratio, neutrophil count, total lymphocyte count, total white cell count (WCC), and the levels of total IgG, IgA, and IgM. We examined the lymphocyte percentage and total cell count for each category of lymphocytes. Furthermore, we performed laboratory tests on commonly used clinical biomarkers, including hemoglobin (Hg), platelets, rheumatoid factor (RF), anti-nuclear antibodies (ANA), C-reactive protein (CRP), iron, transferrin, transferrin saturation percentage, ferritin, folate, creatine kinase (CK), free thyroxine (FT4), and thyroid stimulating hormone (TSH). Abnormal values for each marker were determined as patient laboratory values that deviated either above or below the reference levels [51]. All laboratory tests were conducted at The Mater Misericordiae University Hospital, ensuring consistency in testing methodology and minimizing variations in results.

### 4.4. Tools for Processing Data

To manage, analyze, and visualize the collected data, we employed several Python libraries, each serving specific functions to enhance our analysis. The library ‘Pandas’ was crucial for organizing and preprocessing the data, enabling us to structure pain rating and biomarker data [69]. ‘NumPy’ was used for numerical calculations and transformations, providing the computational backbone for our data manipulation tasks [70].

For data visualization, we utilized ‘Matplotlib’ and ‘Seaborn’. Matplotlib allowed us to create various visual representations, such as line plots and bar graphs, which helped illustrate trends and comparisons clearly [71]. Seaborn complemented these visualizations by offering advanced plot types, like violin plots, boxplots, and histograms equipped with kernel distribution estimation [72]. These plots were instrumental in examining the distribution and variability of the pain ratings and biomarker values at each specified time point (T0 and T2).

Statistical analysis was conducted using two-sample Kolmogorov–Smirnov (K-S) and Mann–Whitney U tests. The Kolmogorov–Smirnov (K-S) and Mann–Whitney U tests were used due to the non-parametric nature of the data and the focus on assessing distribution differences. These tests are more appropriate for non-normal data distributions and provide a robust analysis less affected by outliers and skewed data [73,74,75]. The K-S test assessed the dissimilarity between the distributions of the two groups, providing a statistic indicating this dissimilarity’s extent. A higher K-S statistic signifies a more significant divergence between the time points. The Mann–Whitney U test complemented this by evaluating differences in the central tendency—mean or median—between the two time points, T0 and T2. The significance of findings from both tests was determined by *p*-values, with thresholds set at less than 0.05, 0.01, or 0.001 to indicate statistical significance. Smaller *p*-values represented more substantial evidence against the null hypothesis, suggesting significant differences between time points.

We developed cluster maps to analyze the temporal changes in the relationships between pain ratings and statistically relevant biomarkers. Furthermore, we employed the K-nearest neighbors (KNN) technique to compute the Euclidean distances between individual pain clusters (1–10) and their associated biomarker values. Patients were partitioned into ten clusters based on their reported pain levels, ranging from 1 to 10. Using the KNN technique, we calculated the median Euclidean distances between patients’ relevant biomarkers within the same pain cluster at T0 and T2. These Euclidean distances were normalized to absolute values and considered unitless.

Further, we employed a robust linear model (RLM) using iteratively reweighted least squares (IRLS) with and without the constant term to explore the impact of various biomarkers on changes in pain ratings over time. This model, particularly resistant to outliers, used the Huber T norm to minimize the influence of extreme data points and the median absolute deviation for robust scale estimation. We analyzed changes in biomarker values between the time points to determine their predictive power concerning changes in pain. The covariance matrix of the estimators was computed using the type H1 estimator to accommodate heteroscedasticity, thus ensuring the validity of the standard errors and test statistics.

We employed multiple software tools to analyze data. To streamline our study, we utilized Microsoft Excel version 16.8 to arrange, classify, and examine patient pain and laboratory test results.

## 5. Conclusions

Our study involving 186 patients with PTLDS/TBIs revealed several critical insights into the relationship between demographic factors, pain ratings, and biomarkers before and after antibiotic treatment. We found no significant influence of age or gender on pain ratings, suggesting that these demographic factors do not affect the pain experienced within our study population. However, there was a notable reduction in pain ratings following antibiotic treatment. This change in pain scores over time in patients receiving combination antibiotic therapy for their chronic Lyme and other TBIs suggests the presence of a persistent infection that responded to antibiotic treatment. Despite significant changes in the distribution and median values of transferrin, CD4%, platelets, neutrophils, and transferrin saturation %, their correlation with pain ratings was only significant within a closed model, not allowing for non-measured biomarkers. This indicates that while these biomarkers may have clinical importance, they do not directly relate to the experience of pain reported by patients. Patients reporting the same pain level had the closest cluster biomarker profiles, indicating consistency in their pain reporting. After treatment, however, for patients reporting a pain level of 10, their biomarker profiles became more similar to those of patients with lower pain ratings. This may indicate a normalization progression. These results highlight issues in human rights regarding withholding pain relief for patients with TBIs by withholding antibiotic treatment. However, it is important to acknowledge that opinions differ on managing chronic Lyme disease. For example, the IDSA guidelines recommend against additional antibiotic therapy for patients with persistent symptoms following recommended treatment without objective evidence of reinfection or treatment failure. Our findings suggest that further investigation is warranted to reconcile these differing perspectives and to develop evidence-based guidelines that ensure optimal patient care.

## Figures and Tables

**Figure 1 antibiotics-13-00693-f001:**
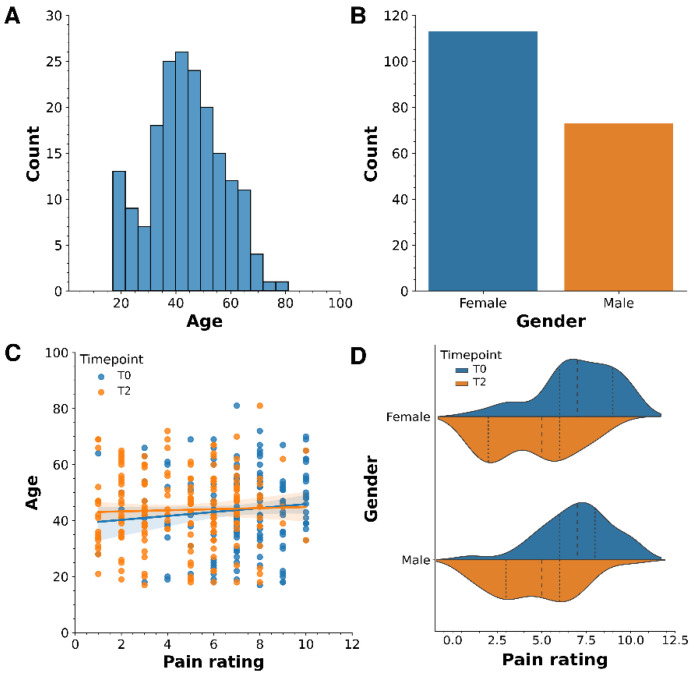
Age and gender do not influence pain ratings at time points T0 and T2. The charts in panels (**A**,**B**) show the overall distribution of age and gender in the study. Panels (**C**,**D**) show the distribution of pain ratings across different ages and genders, respectively, at time points T0 and T2.

**Figure 2 antibiotics-13-00693-f002:**
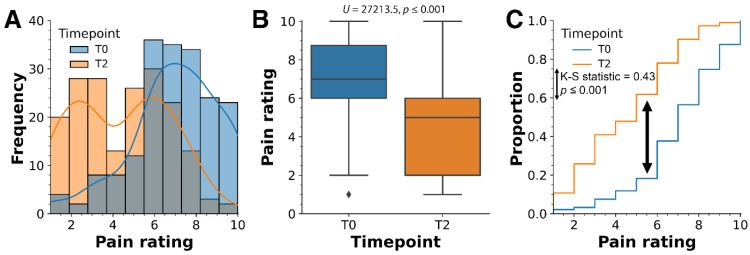
The pain ratings have significantly decreased from time point T0 to T2. Panel (**A**) displays the pain ratings recorded at these time points. Panel (**B**) illustrates a boxplot showing a decrease in the median pain ratings from T0 to T2. The statistical analysis from the Mann–Whitney U test is also included in this panel. Panel (**C**) presents the empirical distribution function (ECDF) of pain ratings at T0 and T2, including the results from the Kolmogorov–Smirnov (K-S) test.

**Figure 3 antibiotics-13-00693-f003:**
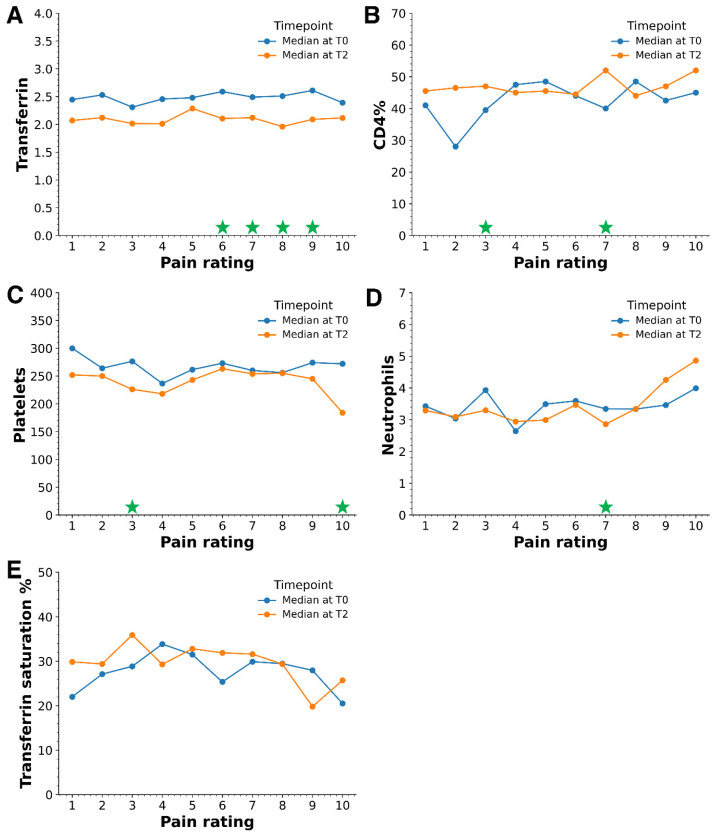
The median transferrin levels are consistently lower at time point T2 across all pain ratings, with notable decreases observed, particularly from pain ratings 6 to 9. The panel compares the median values at time points T0 and T2 for (**A**) transferrin, (**B**) CD4%, (**C**) platelets, (**D**) neutrophils, and (**E**) transferrin saturation %. Green stars on the *x*-axis indicate the pain ratings where differences in median values were statistically significant (*p*-value ≤ 0.05). For a detailed statistical analysis, refer to Appendix A.

**Figure 4 antibiotics-13-00693-f004:**
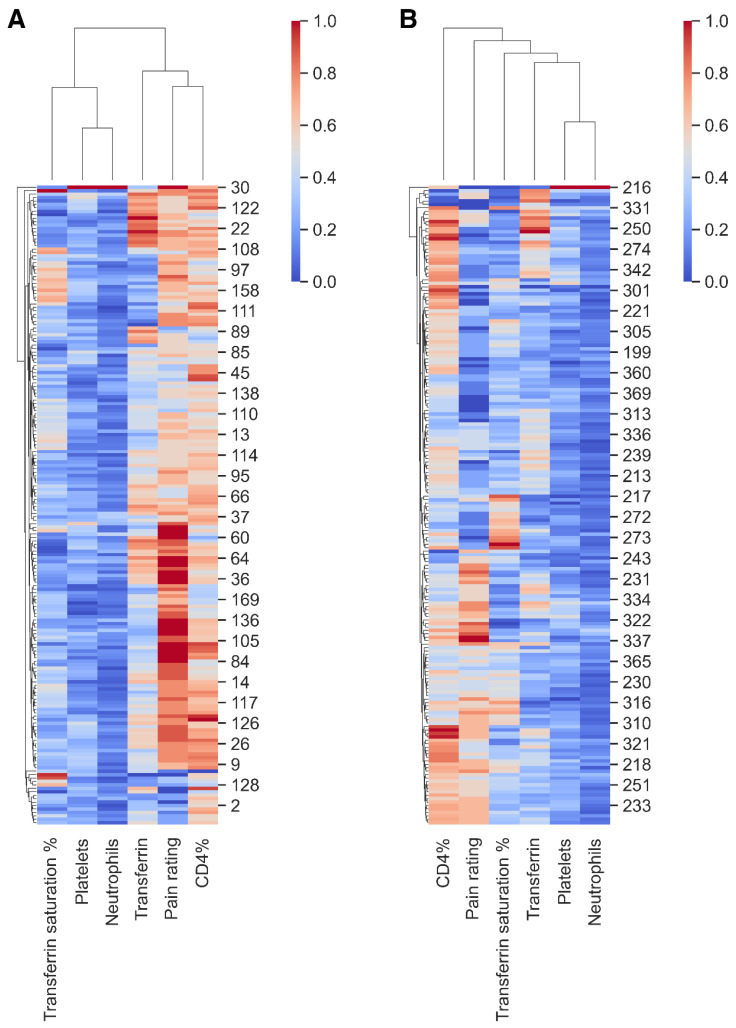
Temporal analysis reveals correlations between pain ratings and biomarkers (transferrin, CD4%, platelets, neutrophils, transferrin saturation %), suggesting external influences on pain perception. The panel displays cluster maps at time points: (**A**) before treatment (T0) and (**B**) after treatment (T2). Each map features a top dendrogram that clusters the pain ratings with biomarkers and a left dendrogram that groups patients based on their similarities. Blue and red colors within each cluster map denote lower and higher values, respectively. For detailed Euclidean distances of individual pain ratings, refer to Appendix A.

**Table 1 antibiotics-13-00693-t001:** Significant differences in distribution and median values of transferrin, transferrin saturation (%), platelets, and CD4% between time points T0 and T2. This table details the statistical analysis of Mann–Whitney U and Kolmogorov–Smirnov (K-S) and tests to compare each biomarker’s median and distribution values at the two time points. For visual representation, refer to Appendix A, which show histograms, empirical cumulative distribution functions (ECDFs), and boxplots for each biomarker.

Biomarker (Reference Range)	Median		Mann–Whitney U Test Statistic	Kolmogorov–Smirnov (K-S) Test Statistic
	T0	T2		
Transferrin (1.88–3.02 g/dL)	2.50	2.09	25,815 ***	0.40 ***
Transferrin saturation (%) (19–55%)	27.8	31.6	13,646 ***	0.20 ***
Platelets (150–400 × 10^9^/L)	269	245	20,555 **	0.19 **
CD4% (32–60%)	44	47	14,399 **	0.15 *
Neutrophils (2–8 × 10^9^/L)	3.4	3.1	19,944 *	0.15 *
CD4+ Helper T-cell count (540–1600 cells/μL)	822	914	15,593	0.13
Hemoglobin (11.5–16.5 g/dL)	14.1	13.9	18,633	0.11
Helper/suppressor (H/S) ratio (0.9–4.5)	2.07	2.08	15,883	0.09
CD3% (61–84%)	70	71	15,927	0.09
White cell count (3.5–11 × 10^9^/L)	6.1	5.8	19,070	0.08
CD3 Total (960–2600 cells/μL)	1304	1420	16,421	0.08
Lymphocytes (1–4 × 10^9^/L)	1.8	1.8	16,558	0.06
Ferritin (22–275 μg/mL)	76	75	18,076	0.06
Creatine kinase (33–208 I.U./L)	81	81	16,861	0.06
Iron (6–33 μmol/L)	17.8	17.6	17,623	0.05
CD8% (13–40%)	22	22	17,350	0.05
CD8-suppressor count (270–930 cells/μL)	419	426	17,203	0.04
C-reactive protein (<7 mg/L)	1	1	17,514	0.03

* *p*-value ≤ 0.05, ** *p*-value ≤ 0.01, *** *p*-value ≤ 0.001.

**Table 2 antibiotics-13-00693-t002:** The robust linear regression model without a constant demonstrates that variations in transferrin and CD4% between T0 and T2 significantly predict pain changes.

Variable	Coefficient	Standard Error	Z-Statistic	*p* Value	95% Confidence Interval
Constant	NA	NA	NA	NA	NA
Transferrin	3.29	0.54	6.05	0.000	2.22–4.36
CD4%	−0.12	0.03	−3.15	0.002	−0.20–−0.04
Platelets	0.006	0.005	1.28	0.19	−0.004–0.01
Neutrophils	−0.07	0.16	−0.47	0.63	−0.40–0.24
Transferrin saturation %	0.003	0.01	0.21	0.83	−0.02–0.03

NA = Not applicable.

**Table 3 antibiotics-13-00693-t003:** Robust linear regression model analysis with a constant indicates that variations in transferrin, CD4%, platelets, neutrophils, and transferrin saturation % between time point T0 and T2 do not significantly predict changes in pain, with only the model’s constant, indicating a significant decrease. The constant represents the average reduction in pain when biomarker levels are held constant at zero, highlighting a general decrease in pain across the sample.

Variable	Coefficient	Standard Error	Z-Statistic	*p* Value	95% Confidence Interval
Constant	−2.49	0.26	−9.37	0.000	−3.01–−1.97
Transferrin	−0.09	0.58	−0.16	0.86	−1.24–1.04
CD4%	−0.02	0.03	−0.72	0.47	−0.09–0.04
Platelets	−0.004	0.005	−0.99	0.31	−0.01–0.005
Neutrophils	0.02	0.14	0.14	0.88	−0.26–0.30
Transferrin saturation %	−0.01	0.01	−0.99	0.31	−0.04–0.01

## Data Availability

The datasets generated and analyzed during the current study are available in the GitHub repository. The data are provided in CSV format, and the Jupyter notebook used for the analysis is also included to ensure complete transparency and reproducibility of the study’s findings. These resources can be accessed at https://github.com/kugarg/Pain-Biomarker-TBI.git (accessed on 24 July 2024). Researchers and interested parties are encouraged to review and utilize the data and analysis code under the repository’s licensing terms, which support open access and collaborative improvements. If there are any questions or requests for additional information, please contact the corresponding author.

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
