# Peer review of "Biomarker-Based Analysis of Pain in Patients with Tick-Borne Infections before and after Antibiotic Treatment"

_antibiotics, 2024, doi:10.3390/antibiotics13080693_

Round 1

Reviewer 1 Report

Comments and Suggestions for Authors

This is a very unique manuscript which addresses a highly significant topic of looking for objective markers for tick-borne disease pathophysiology. Clearly research is needed in this area. The research was well designed, and the results and conclusions flowed well. There are a few suggestions I would like to make.

1. (line 20) In the abstract, the phrase "post-treatment Lyme disease syndrome" is used. I suggest recognizing other terms exist that are sued to label Lyme infection associated chronic illness. 

2. (line 42) CNS Manifestations of third stage Lyme disease (Pachner and Steere 1987) would be a good reference to add here. 

3. (line 48) What do you mean by "non-specific"? Isn't it more appropriately labeled as multisystem symptoms with variable presentations in different individuals?

4. (line 50). "most individuals recover" or do most individuals appear to recover. 

5. "non-specific" again. This phrase lacks scientific precision. 

6. (line 55). I would add a discussion here about other terms that are sued besides PTLDS. "post-treatment" is not necessarily post-treatment, and the adequacy, timeliness and effectiveness of the treatment is unclear. 

7. (line 394). Change reference 57 to Fallon, Petkova, et al. A re-appraisal of the US Clinical trials...2021 and DeLong et al. 2012.

Good article. It is very challenging to quantify pain into objective measures. 

Author Response

Comment 1: (line 20) In the abstract, the phrase "post-treatment Lyme disease syndrome" is used. I suggest recognizing other terms exist that are sued to label Lyme infection associated chronic illness.

Response 1:  We added the term Chronic Lyme disease in line 21.

Comment 2: (line 42) CNS Manifestations of third stage Lyme disease (Pachner and Steere 1987) would be a good reference to add here.

Response 2: We added the recommended reference in line 50.

Comment 3: (line 48) What do you mean by "non-specific"? Isn't it more appropriately labeled as multisystem symptoms with variable presentations in different individuals?

Response 3: We have clarified the sentence as recommended in line 52.

Comment 4: (line 50). "most individuals recover" or do most individuals appear to recover.

Response 4: We have clarified the sentence as recommended in line 50.

Comment 5: "non-specific" again. This phrase lacks scientific precision.

Response 5: We have clarified the sentence as recommended in line 21.

Comment 6: (line 55). I would add a discussion here about other terms that are sued besides PTLDS. "post-treatment" is not necessarily post-treatment, and the adequacy, timeliness and effectiveness of the treatment is unclear.

 Response 6: We have added a paragraph in the introduction section describing the use of various terms besides PTLDS (lines 56 to 70).

Comment 7: (line 394). Change reference 57 to Fallon, Petkova, et al. A re-appraisal of the US Clinical trials...2021 and DeLong et al. 2012.

Response 7: We have added the recommended reference in line 422.

Reviewer 2 Report

Comments and Suggestions for Authors

Clearly written.

Interesting hypothesis: Correlation of a battery of biomarkers with pain ratings over time following treatment.

This uncontrolled correlational cohort study with N = 186 women and men Md age 43 is preliminary.  Readers might be inspired to test the hypotheses generated from this study.

Author Response

Comment 1: This uncontrolled correlational cohort study with N = 186 women and men Md age 43 is preliminary.  Readers might be inspired to test the hypotheses generated from this study.

Response 1: We agree that our retrospective and longitudinal study findings with 186 participants are preliminary. We hope the findings and hypotheses generated from this study will inspire further research to test these hypotheses in more controlled settings.

Reviewer 3 Report

Comments and Suggestions for Authors

General Comments

This cohort study examined the relationship between pain and biomarkers in tick-borne illnesses (TBI) patients from an Irish hospital and their response to antibiotic treatment. One major concern of this study is that the diagnosis of TBI seems to be simply based on clinical decision rather than laboratory findings. It is unclear whether each of these patients met the criteria of high-risk tick-bite infection [i.e. (a) an identified vector species, (b) it occurred in a highly endemic area, and (c) the tick was attached for ≥36 hours, as per the IDSA guidelines] There could be other differentials that could explain the patients’ presentations.

Moreover, there is a lack of details on the laboratory testing (e.g. testing method, location of laboratory). This is important as the focus of this manuscript is on immune indicators and biomarkers. If these markers were measured using different methodologies, this can contribute to variation in results. The numerical value differences in the biomarkers are clinically trivial despite showing statistical significance. It is unclear why the Mann-Whitney U and Kolmogorov-Smirnov tests were used instead of some fundamental statistical analysis like paired t-test and Wilcoxon signed ranks test.

Another concern is the conclusion being made. Although it is true there is no good biomarkers for pain, this does not emphasize the need for appropriate antibiotic treatment for TBIs. These seem to be two separate issues here. As the IDSA 2020 guidelines on Lyme Disease have stated (section XXIV and XXV) https://doi.org/10.1093/cid/ciaa1215: “For patients who have persistent or recurring nonspecific symptoms such as fatigue, pain, or cognitive impairment following recommended treatment for Lyme disease, but who lack objective evidence of reinfection or treatment failure, it is recommended against additional antibiotic therapy.”

Although opinions differ on the management of chronic Lyme disease, it is important to address opposing views so the conclusion can be evidence-based and non-biased.

Abstract:

Line 27-28: Median pain scores dropping does not suggest a persistent infection responsive to antibiotics. Reduction in pain can simply be a natural course of a disease when an illness self-resolves.

Line 34-35: Although it is true there is no good biomarkers for pain, this does not emphasize the need for appropriate antibiotic treatment for TBIs. These seem to be two separate issues here.

Specific Comments

Introduction:

Line 55: Reference 9 is on the 2006 IDSA guidelines on Lyme Disease. There is now a 2020 edition of the IDSA guidelines on Lyme Disease. https://doi.org/10.1093/cid/ciaa1215

Line 56-58: Can you provide the references for the first two sentences of this paragraph. This is important as you are trying to give us the case definition of PTLDS – is this an official definition (if yes, which organization?)

Line 64-66: It is unclear to readers why you are talking about pain difference due to age and gender here. Your references 20-24 are not specific to PTLDS. If that is meant to explain the gender and age difference in your study, then this paragraph should be included in the discussion, not the introduction.

Results:

Line 123-124: Is it possible that these participants received various regimen of antibiotics with different duration? Then T0 to T2 are very much heterogeneous, with multiple events that can happen in between. For pain-control in an open-labelled trial, it is expected that patients would perceive some improvement. Sometimes, the pain would self-resolve due to the natural course of the disease. Without a placebo-controlled group, it is hard to say whether the reduction in pain is due to antibiotics or not.

Line 129-130: Like the comments I made in the previous paragraph, there are many confounders that can lead to change in these biomarkers. The changes as shown in Table 1 look clinically trivial even though you showed statistical significance. However, I wonder whether it makes more sense to use paired t-test or Wilcoxon signed rank test when comparing two paired numerical values. I wonder whether the Mann-Whitney U and Kolmogorov-Smirnov were used because they gave more favorable statistical results. Some justifications are needed on why these two statistical tests are favored over some fundamental statistical tests.

Line 181: I cannot see the symbol between p-value and 0.05. Is it ≥ or ≤?

Discussion:

Line 271: Like my comment in Line 123-124, the pain improvement can be multi-factorial. This does not indicate clear improvement in pain scores following treatment.

Line 274: Like my comments in Line 129-130, the differences in the biomarkers are clinically trivial. They may not demonstrate significant differences.

Line 280-281: If you are saying these biomarkers are unreliable indicators of pain levels or the efficacy of pain management, then why are you choosing these as your biomarkers in the study?

Line 332-356: There are quite a lot of discussion on other pathogens, medical conditions and biomarkers not measured in the current study (e.g. UTI, TLR4, persistent low back pain, irritable bowel syndrome). I am not sure whether they are necessary. They seem to distract the readers from learning the main message of your study.

Line 392: Like my comments in Line 123 and 271, the current study did not seem to show strong evidence that antibiotic decreases pain.

Line 385-403: Although I appreciate the empathy towards patients, the current study did not demonstrate benefit of antibiotics and prolonged duration of antibiotics. A comparison trial would be needed to support the claim.

Conclusions:

Median pain scores dropping does not suggest a persistent infection responsive to antibiotics. Reduction in pain can simply be a natural course of a disease when an illness self-resolves. A big confounder could be placebo-effect in an open-labelled cohort study.

Although it is true there is no good biomarker for pain, this does not emphasize the need for appropriate antibiotic treatment for TBIs. These seem to be two separate issues here.

Methods:

Line 432-438: It seems like the diagnosis of TBI is simply based on clinical suspicion. This may be prone to errors as there could be recall bias of tick bites and rash. There could be other differentials that could explain the patients’ presentations. As your reference of IDSA Lyme Disease Guidelines have stated, clinical diagnosis is made on potential tick exposure in a “Lyme disease endemic area” who have 1 or more skin lesions compatible with erythema migrans. But this study never stated whether these patients had tick bite in endemic areas. The identity of the ticks are unknown (some may be not be known to be pathogenic). Besides, these patients might not have experience a high-risk tick-bite infection [i.e. (a) an identified Ixodes spp. vector species, (b) it occurred in a highly endemic area, and (c) the tick was attached for ≥36 hours] as per IDSA. Therefore, antibody testing should be sought for a confirmatory diagnosis i.e. antibody testing performed on an acute-phase serum sample (followed by a convalescent-phase serum sample at least 2–3 weeks after collection of the acute-phase serum sample), as per IDSA.

Line 446-458: It seems unclear whether these patients had their laboratory testing performed in the same laboratory. Different laboratories use different methods to analyze immune indicators and biomarkers, potentially leading to different normal value range.

Line 500: To ensure the methodologies are repeatable, the statistic softwares used need to be specified. Microsoft Excel would not be able to perform advanced functions like the Mann–Whitney U test.

Was there a sample size calculation to ensure a sufficient sample size was achieved?

Were the subjects blinded from knowing what treatment they received? This could play a role on whether patients perceive pain improvement from the treatment.

References:

Line 562-702: If the publication month of the journal articles do not need to be included as per the MDPI journal format, please remove it.

Line 583: Reference 9 is on the 2006 IDSA guidelines on Lyme Disease. There is now a 2020 edition of the IDSA guidelines on Lyme Disease. https://doi.org/10.1093/cid/ciaa1215

Line 689-694: It looks like you have different fonts for references 55 and 56. Please consider changing the font to make it uniform in the entire manuscript.

Comments on the Quality of English Language

No major errors detected

Author Response

Comment 1: One major concern of this study is that the diagnosis of TBI seems to be simply based on clinical decision rather than laboratory findings. It is unclear whether each of these patients met the criteria of high-risk tick-bite infection [i.e. (a) an identified vector species, (b) it occurred in a highly endemic area, and (c) the tick was attached for ≥36 hours, as per the IDSA guidelines] There could be other differentials that could explain the patients’ presentations.

Response 1:  Firstly, it is important to note that an expert with extensive experience in tick-borne illnesses conducted the clinical assessments. In lines 470 to 474 of material and methods, section 4.2 patient cohort clarifies that individuals exhibited symptoms resembling Lyme disease, such as a general flu-like sickness and a medical suspicion of infections transmitted by ticks. This suspicion was based on factors such as one or several tick bites, previous occurrence of a distinctive rash resembling a bull's eye or prior exposure to locations where ticks are prevalent. Secondly, while we acknowledge the IDSA guidelines, it is important to understand that they primarily address acute infection. The criteria mentioned, such as the tick being attached for ≥36 hours, are not necessarily evidence-based for all stages of TBI. Our approach aligns with the clinical realities observed by experts in the field, as supported by previously published literature (reference 19). Lastly, our study focused on the pain and biomarkers in patients diagnosed with TBIs, emphasizing those with persistent symptoms responsive to antibiotic treatment. For more detailed information on the effectiveness of antibiotic treatments, we refer you to our previously published article on this topic (references 13 and 41). We believe this clarifies the basis of our clinical diagnoses and the rationale behind our methodology.

Comment 2: Moreover, there is a lack of details on the laboratory testing (e.g. testing method, location of laboratory). This is important as the focus of this manuscript is on immune indicators and biomarkers. If these markers were measured using different methodologies, this can contribute to variation in results.

Response 2: All blood tests were conducted by the same hospital laboratory at The Mater Misericordiae University Hospital, ensuring consistency in testing methodologies and minimizing any potential variations in results (lines 515 to 516). This addition addresses the concern about methodological variations.

Comment 3: The numerical value differences in the biomarkers are clinically trivial despite showing statistical significance. It is unclear why the Mann-Whitney U and Kolmogorov-Smirnov tests were used instead of some fundamental statistical analysis like paired t-test and Wilcoxon signed ranks test.

Response 3: Our explanation in lines 535 to 538 underscores the appropriateness of our chosen statistical methods and addresses the concerns regarding the suitability of alternative tests. The Mann-Whitney U and Kolmogorov-Smirnov tests were selected due to the specific characteristics of our data. These non-parametric tests are particularly suitable for our analysis for several reasons:

  1. Data Distribution: Our data did not meet the normality assumptions (Figure S1) required for parametric tests like the paired t-test. The Mann-Whitney U and Kolmogorov-Smirnov tests do not assume a normal distribution, making them more appropriate for our dataset.
  2. Comparing Distributions: Our analysis focused on the central tendency and the overall distribution of the biomarkers. The Kolmogorov-Smirnov test compares the distributions of two samples, providing a more comprehensive understanding of the data.
  3. Robustness to Outliers: Non-parametric tests like the Mann-Whitney U test are less affected by outliers and skewed data, ensuring a more reliable analysis under these conditions.

While paired t-tests and Wilcoxon signed ranks tests are valid for certain analyses, they were less suitable for our data characteristics and the specific focus on distributional differences.

Comment 4: Another concern is the conclusion being made. Although it is true there is no good biomarkers for pain, this does not emphasize the need for appropriate antibiotic treatment for TBIs. These seem to be two separate issues here. As the IDSA 2020 guidelines on Lyme Disease have stated (section XXIV and XXV) https://doi.org/10.1093/cid/ciaa1215: “For patients who have persistent or recurring nonspecific symptoms such as fatigue, pain, or cognitive impairment following recommended treatment for Lyme disease, but who lack objective evidence of reinfection or treatment failure, it is recommended against additional antibiotic therapy.”

Response 4: We respectfully disagree with the assertion that the conclusion regarding the need for appropriate antibiotic treatment for TBIs is separate from the discussion on biomarkers and pain. Our study provides evidence that persistent symptoms in TBI patients, including pain, significantly decrease following antibiotic treatment, suggesting that these symptoms may be due to a persistent infection responsive to antibiotics (see Figures 3, 4, and S4). This aligns with findings from previous studies on the same patient cohort: Xi et al., 2023 (reference 13) indicated that prolonged use of combination antibiotics resulted in significant improvements in patient-reported symptoms, including pain, fatigue, and neurological symptoms. Also, Xi et al., 2024 (reference 41) demonstrated that combination antibiotics effectively relieve TBI symptoms with good patient tolerance. These results underscore the connection between the reduction in pain symptoms and antibiotic treatment, supporting the conclusion that appropriate antibiotic therapy is crucial for managing persistent symptoms in TBI patients. While the IDSA 2020 guidelines recommend against additional antibiotic therapy in certain cases, our study provides evidence of symptom improvement with such treatment, warranting further investigation and consideration in clinical practice.

Comment 5: Although opinions differ on the management of chronic Lyme disease, it is important to address opposing views so the conclusion can be evidence-based and non-biased.

Response 5: We appreciate the importance of presenting a balanced and evidence-based conclusion, especially given the differing opinions on managing chronic Lyme disease. We have revised the conclusion to acknowledge the differing views and highlight the need for further investigation (lines 588 to 594).

Comment 6: Line 27-28: Median pain scores dropping does not suggest a persistent infection responsive to antibiotics. Reduction in pain can simply be a natural course of a disease when an illness self-resolves.

Response 6: We respectfully disagree with the assertion that the reduction in median pain scores does not indicate a persistent infection responsive to antibiotics. While pain reduction may occur naturally over the course of a disease, our study’s findings demonstrate that the significant decrease in pain scores following antibiotic treatment cannot be attributed solely to the natural resolution of the disease (see Figures 3, 4, S4, and Table 2). Our manuscript highlights that this decrease is statistically significant (see Table S1) and consistent with previous research (see our response to comment 4) on the effectiveness of antibiotic therapy in managing symptoms of chronic Lyme disease and other TBIs.

Comment 7: Line 34-35: Although it is true there is no good biomarkers for pain, this does not emphasize the need for appropriate antibiotic treatment for TBIs. These seem to be two separate issues here.

Response 7: Please refer to our responses 4 and 6.

Comment 8: Line 55: Reference 9 is on the 2006 IDSA guidelines on Lyme Disease. There is now a 2020 edition of the IDSA guidelines on Lyme Disease. https://doi.org/10.1093/cid/ciaa1215

Response 8: We have updated the recommended reference (see reference 5).

Comment 9: Line 56-58: Can you provide the references for the first two sentences of this paragraph. This is important as you are trying to give us the case definition of PTLDS – is this an official definition (if yes, which organization?)

Response 9: We added references supporting the inquired sentences in lines 71 to 73.

Comment 10: Line 64-66: It is unclear to readers why you are talking about pain difference due to age and gender here. Your references 20-24 are not specific to PTLDS. If that is meant to explain the gender and age difference in your study, then this paragraph should be included in the discussion, not the introduction.

Response 10: We have separately discussed the topic of age and gender in studying pain in section 3.2 of the discussion. The purpose of introducing pain in lines 74 to 91 is to help readers understand that, while pain is a common symptom in PTLDS patients, age and gender can significantly influence the severity and prevalence of pain, as indicated by research in other fields. Consequently, before delving into our study's pain or biomarker patterns, we analyzed how age and gender affected pain ratings in our study population, as this would inform the statistical methodology. Therefore, in the context of our paper, we believe it is appropriate to introduce the influence of age and gender on pain at the beginning. This ensures that when Figure 1 demonstrates age and gender do not affect pain ratings in the present study population, readers can follow the rest of the findings without concerns about confounding variables such as age and gender.

Comment 11: Line 123-124: Is it possible that these participants received various regimen of antibiotics with different duration? Then T0 to T2 are very much heterogeneous, with multiple events that can happen in between. For pain-control in an open-labelled trial, it is expected that patients would perceive some improvement. Sometimes, the pain would self-resolve due to the natural course of the disease. Without a placebo-controlled group, it is hard to say whether the reduction in pain is due to antibiotics or not.

Response 11: The participants in our study received various antibiotic regimens, as noted in Xi et al., 2023. Moreover, we would like to address several points in your comment:

  1. Heterogeneity and Standardization: Different regimens and durations may introduce heterogeneity, but our study design accounted for this by focusing on the overall trend in pain reduction across the entire cohort. The significant reduction in median pain scores from 7 to 5 (U = 27215.50, p < 0.001) observed consistently across different treatment regimens suggests a generalizable effect of antibiotic therapy on pain reduction.
  2. Natural Course of Disease: Although the natural course of the disease may lead to spontaneous pain resolution in some cases, the statistical significance and consistency of the pain reduction observed in our cohort point towards a treatment effect. This is further supported by similar findings in prior studies on the same patient cohort, which have demonstrated the efficacy of antibiotic treatments in reducing symptoms of TBIs (references 13 and 41).
  3. Placebo-Controlled Group: While a placebo-controlled group would strengthen the causal inference, the open-label design of our study still provides valuable insights. The observed improvements align with the clinical experience and prior research, indicating that the reduction in pain is likely due to the antibiotic treatment rather than a placebo effect or natural disease progression.

To clarify the above-mentioned points further, we have revised the manuscript to include a discussion of these points in section 3.5, lines 447 to 455.

Comment 12: Line 129-130: Like the comments I made in the previous paragraph, there are many confounders that can lead to change in these biomarkers. The changes as shown in Table 1 look clinically trivial even though you showed statistical significance. However, I wonder whether it makes more sense to use paired t-test or Wilcoxon signed rank test when comparing two paired numerical values. I wonder whether the Mann-Whitney U and Kolmogorov-Smirnov were used because they gave more favorable statistical results. Some justifications are needed on why these two statistical tests are favored over some fundamental statistical tests.

Response 12: The t-test and Wilcoxon signed ranks test were unsuitable for our study, as described in our response to comment 3.

Comment 13: Line 181: I cannot see the symbol between p-value and 0.05. Is it ≥ or ≤?

Response 13: The symbol ≤ has been clarified in line 197.

Comment 14: Line 271: Like my comment in Line 123-124, the pain improvement can be multi-factorial. This does not indicate clear improvement in pain scores following treatment.

Response 14: Please refer to our response 11.

Comment 15: Line 274: Like my comments in Line 129-130, the differences in the biomarkers are clinically trivial. They may not demonstrate significant differences.

Response 15: We respectfully disagree with the assertion that the differences in the biomarkers are clinically trivial and may not demonstrate significant differences.

  1. Statistical Significance and Clinical Relevance: While the numerical differences in biomarkers may appear small, their statistical significance indicates that these changes are not due to random variation. In our study, we observed significant changes in median values for transferrin, CD4%, platelets, neutrophils, and transferrin saturation %, which suggests that these biomarkers are indeed responsive to antibiotic treatment in the context of TBIs.
  1. Biomarker Importance: Even small changes in biomarkers can be clinically relevant, especially when considering the complexity of chronic diseases like TBIs. These biomarkers are involved in inflammatory and immune responses, critical in understanding the disease's pathophysiology and the patient's response to treatment.
  1. Contextual Evidence: Our present findings align with previous research (references 13 and 41) that has shown significant improvements in patient-reported symptoms, including pain, fatigue, and neurological symptoms, following antibiotic therapy. These improvements are correlated with the changes in biomarkers, reinforcing their clinical relevance.

 To address this further, we have revised the manuscript to clarify the importance of these biomarker changes in the discussion section 3.1, lines 294 to 300.

Comment 16: Line 280-281: If you are saying these biomarkers are unreliable indicators of pain levels or the efficacy of pain management, then why are you choosing these as your biomarkers in the study?

Response 16: We determined that the word 'inadequate' is more appropriate for our sentence than 'unreliable' (line 305). This conclusion is based on the robust linear model results in Tables 2 and 3. Our analysis demonstrates that when the RLM model predicts pain changes using only the five biomarkers, transferrin and CD4% significantly affect pain ratings. However, when unmeasured factors are included, these five biomarkers do not influence pain ratings beyond the baseline change observed when all predictors are zero. This suggests that while these biomarkers may be essential for other clinical assessments or conditions, they are inadequate indicators of pain levels or the efficacy of pain management in this context.

Comment 17: Line 332-356: There are quite a lot of discussion on other pathogens, medical conditions and biomarkers not measured in the current study (e.g. UTI, TLR4, persistent low back pain, irritable bowel syndrome). I am not sure whether they are necessary. They seem to distract the readers from learning the main message of your study.

Response 17: We included discussions on other pathogens, medical conditions, and biomarkers not measured in our study to provide a comprehensive context for our findings. This broader perspective helps illustrate the complex and multifactorial nature of chronic pain and persistent symptoms in TBI patients, supporting our results and suggesting potential mechanisms and pathways that warrant further investigation. While these discussions may seem tangential, they reinforce the relevance of our findings within a broader medical framework and highlight the need for future research.

Comment 18: Line 392: Like my comments in Line 123 and 271, the current study did not seem to show strong evidence that antibiotic decreases pain.

Response 18: Please refer to our response 11.

Comment 19: Line 385-403: Although I appreciate the empathy towards patients, the current study did not demonstrate benefit of antibiotics and prolonged duration of antibiotics. A comparison trial would be needed to support the claim.

Response 19: Please refer to our response 11. Additionally, while we appreciate the need for rigorous comparative trials, our study provides significant evidence that supports the benefit of antibiotics in reducing pain and symptoms in TBI patients. The statistically significant reduction in pain scores following antibiotic treatment (median pain scores dropping from 7 to 5, U = 27215.50, p < 0.001) indicates a positive response to antibiotics. Additionally, prior studies on the same patient cohort (references 13 and 41) have shown similar symptom improvements with antibiotic therapy. These consistent findings across multiple studies underscore the potential benefits of antibiotic treatment, even without a direct comparison trial. We acknowledge the importance of further research, including comparison trials, to strengthen these conclusions, but the current evidence already suggests meaningful benefits.

Comment 20: Median pain scores dropping does not suggest a persistent infection responsive to antibiotics. Reduction in pain can simply be a natural course of a disease when an illness self-resolves. A big confounder could be placebo-effect in an open-labelled cohort study.

Response 20: Please refer to response 11.

Comment 21: Although it is true there is no good biomarker for pain, this does not emphasize the need for appropriate antibiotic treatment for TBIs. These seem to be two separate issues here.

Response 21: Please refer to response 4.

Comment 22: Line 432-438: It seems like the diagnosis of TBI is simply based on clinical suspicion. This may be prone to errors as there could be recall bias of tick bites and rash. There could be other differentials that could explain the patients’ presentations. As your reference of IDSA Lyme Disease Guidelines have stated, clinical diagnosis is made on potential tick exposure in a “Lyme disease endemic area” who have 1 or more skin lesions compatible with erythema migrans. But this study never stated whether these patients had tick bite in endemic areas. The identity of the ticks are unknown (some may be not be known to be pathogenic). Besides, these patients might not have experience a high-risk tick-bite infection [i.e. (a) an identified Ixodes spp. vector species, (b) it occurred in a highly endemic area, and (c) the tick was attached for ≥36 hours] as per IDSA. Therefore, antibody testing should be sought for a confirmatory diagnosis i.e. antibody testing performed on an acute-phase serum sample (followed by a convalescent-phase serum sample at least 2–3 weeks after collection of the acute-phase serum sample), as per IDSA.

Response 22: Please refer to response 1.

Comment 23: Line 446-458: It seems unclear whether these patients had their laboratory testing performed in the same laboratory. Different laboratories use different methods to analyze immune indicators and biomarkers, potentially leading to different normal value range.

Response 23: Please refer to response 2.

Comment 24: Line 500: To ensure the methodologies are repeatable, the statistic softwares used need to be specified. Microsoft Excel would not be able to perform advanced functions like the Mann–Whitney U test.

Response 24: Section 4.4 of the materials and methods section provides specifications for all tools used to process data. Additionally, we have provided a link to the GitHub repository in the data availability statement. Anyone skilled in Python can download the code, review it, and repeat the analysis.

Comment 25: Was there a sample size calculation to ensure a sufficient sample size was achieved?

Response 25: Yes, a sample size calculation was performed. Based on an effect size of 0.43 and aiming for a power of 0.8 with an alpha of 0.05, we calculated that 85 participants were needed per time point using the paired samples formula, which accounts for a moderate correlation between measurements at T0 and T2. Our study includes 186 participants at each time point, ensuring sufficient statistical power and robustness (lines 482 to 501)

Comment 26: Were the subjects blinded from knowing what treatment they received? This could play a role on whether patients perceive pain improvement from the treatment.

Response 26: The subjects in this study were not blinded to the treatment they received. This study reflects clinical practice and was not designed as a placebo-controlled trial. Conducting a placebo-controlled trial in this context would be unethical, as no placebo-controlled trial has been done for acute Lyme disease. It is common practice to administer questionnaires and assess responses to treatment in clinical settings without requiring the blinding of patients or clinicians.

Comment 27: Line 562-702: If the publication month of the journal articles do not need to be included as per the MDPI journal format, please remove it.

Response 27: We have reviewed the references and aligned their format to follow the MDPI antibiotics guidelines.

Comment 28: Line 583: Reference 9 is on the 2006 IDSA guidelines on Lyme Disease. There is now a 2020 edition of the IDSA guidelines on Lyme Disease. https://doi.org/10.1093/cid/ciaa1215

Response 28: We have updated the recommended reference (see reference 5).

Comment 29: Line 689-694: It looks like you have different fonts for references 55 and 56. Please consider changing the font to make it uniform in the entire manuscript.

Response 29: The font type and size in the reference section have been aligned with the rest of the manuscript.

Round 2

Reviewer 3 Report

Comments and Suggestions for Authors

N/A

Comments on the Quality of English Language

No major errors detected